# The Component Summation Technique for Measuring Upwelling Longwave Irradiance in the Presence of an Obstruction

Bryan E. Fabbri<sup>1,\*</sup>, Gregory L. Schuster<sup>2,5,\*</sup>, Frederick M. Denn<sup>1</sup>, Bing Lin<sup>2</sup>, David A. Rutan<sup>3</sup>, Wenying Su<sup>2</sup>, Zachary A. Eitzen<sup>3</sup>, James J. Madigan Jr.<sup>1</sup>, Robert F. Arduini<sup>4,5</sup>, and Norman G. Loeb<sup>2</sup>

# Abstract.

The CERES Ocean Validation Experiment (COVE) was an instrument suite located at the Chesapeake Light Station approximately 25 kilometers east of Virginia Beach, Virginia (36.9° N, 75.7° W). COVE provided surface verification for the Clouds and the Earth's Radiant Energy System (CERES) satellite measurements for 16 years. However, the large light station occupied approximately 15% of the field of view of the upwelling longwave flux measurement ( $LW^{\uparrow}$ ), so radiation from the structure artificially perturbed the measurements. Hence, we use data from multiple instruments that are not influenced by the structure to accurately obtain  $LW^{\uparrow}$ ; we call this the longwave component summation technique. The instruments required for the component summation are an infrared radiation thermometer to measure sea surface temperature, a pyrgeometer to measure downwelling longwave irradiance, and an air temperature probe. We find a strong negative bias between the obstructed upwelling pyrgeometer measurements and the component summation  $LW^{\uparrow}$  in the colder months, less so in the warmer months. The bias ranged from -6% to +5% over COVE from 2004–2013. These range of biases are larger than the Baseline Surface Radiation Network (BSRN) targeted uncertainties of 2% or 3 W-m<sup>-2</sup> (whichever is greatest), indicating that the component summation technique provides a significant correction to standard BSRN protocols when an obstruction is present. This work documents how we determine the component summation  $LW^{\uparrow}$  irradiances, demonstrates that the calculated values achieve a relative standard error of 0.6% and are within the 2% target uncertainty, and presents guidelines for implementing this methodology at other locations.

Copyright statement. TEXT

<sup>&</sup>lt;sup>1</sup>Analytical Mechanics Associates, Hampton, Virginia, USA

<sup>&</sup>lt;sup>2</sup>NASA Langley Research Center, Hampton, Virginia, USA

<sup>&</sup>lt;sup>3</sup>Adnet Systems Inc., Hampton, Virginia, USA

<sup>&</sup>lt;sup>4</sup>Science Systems and Applications, Inc., Hampton, Virginia, USA

<sup>&</sup>lt;sup>5</sup>Retired

<sup>\*</sup>Corresponding authors: Bryan E. Fabbri, bryan.e.fabbri@nasa.gov and Gregory L. Schuster, 91schuster@gmail.com

## 1 Introduction

Atmospheric irradiance measurements in the shortwave (0.2-5  $\mu$ m) and total (0.2-100  $\mu$ m) channels are essential for characterizing the Earth's radiative energy budget (Wielicki et al., 1996). The Clouds and the Earth's Radiant Energy System (CERES) instruments have collected these vital measurements from multiple satellite platforms since 1997 with a third channel devoted to the window (8-12  $\mu$ m) channel, which was replaced by the longwave band (5-35  $\mu$ m) on the most recent platform (https://ceres.larc.nasa.gov/instruments/). These measurements play a vital role in assessing global temperature changes. CERES directly measures radiation at the top of the atmosphere, and the CERES team uses other measurements and methods to infer the radiation budget in the atmospheric column and at the surface. These level 3 products are not as robust as direct measurements, so the CERES project relies upon direct measurements from surface radiation sites located throughout the world for verification (Kratz et al., 2010).

One of these validation efforts was the CERES Ocean Validation Experiment (COVE), which was a suite of instrumentation that the National Aeronautics and Space Administration (NASA) maintained from 2000-2016 https://science.larc.nasa.gov/CRAVE/COVE/). COVE was located at the Chesapeake Light Station (36.905N, 75.713W), a fixed platform located 25 km east of Virginia (near the mouth of the Chesapeake Bay). The Chesapeake Light Station was built in 1965 as a navigation aid to mark the entrance to the Chesapeake Bay and operated by the United States Coast Guard (USCG; https://www.lighthousefriends.com/light.asp?ID=1691, https://en.wikipedia.org/wiki-/Chesapeake\_Light). NASA and the USCG formalized an inter-agency agreement in 1998 that allowed NASA to use the light station for solar radiation, aerosol, and meteorological measurements. The location and size of the Chesapeake lighthouse provided two characteristics that simplify atmospheric studies from the satellite viewpoint: 1.) a dark surface (the ocean), which simplifies retrievals of the optical properties of aerosols and clouds, and 2.) the lighthouse structure itself is small enough (25x25 meters) that the usual "island effect" associated with oceanic sites is negligible (MISR-Team, 2000; Rutledge et al., 2006).

Many surface radiation sites are networked with other sites that share databases and data acquisition protocols. This includes databases such as the Surface Radiation Budget Network (SURFRAD; https://gml.noaa.gov/grad/surfrad/), the Global Atmosphere Watch Station Information System (GAWSIS; https://gawsis.meteoswiss.ch/GAWSIS), and the World Radiation Monitoring Center (WRMC; https://bsrn.awi.de/project/objectives/) Baseline Surface Radiation Network (BSRN; https://bsrn.awi.de/). For example, COVE measurements adhere to standards set by the BSRN established in 1998 (Ohmura et al., 1998; Driemel et al., 2018). These standards include instrumentation with the highest available accuracy and high temporal resolution (1-3 minutes), near daily cleaning, and rigorous calibration protocols at regular intervals (usually yearly for shortwave and every few years for longwave and meteorological instruments). This renders the BSRN as the most highly-respected archive of long term surface radiation observations in the world, and is why the BSRN is the most commonly used data set for satellite validation of broadband shortwave and longwave radiation (Jin et al., 2003; Rutan et al., 2009, 2015; Kato et al., 2018).

Nearly all of the BSRN sites are located on land, which makes COVE unique as the only true water site in the BSRN database. COVE was located outside the surf zone and far enough away from shore to make it an excellent validation site for space-borne retrievals of cloud and aerosol microphysics. COVE provided better comparisons to satellite measurements than

other scene types like snow, forest, desert, grassland, etc. (Belward and Loveland, 1996, and ceres.larc.nasa.gov/data/general-product-info/). Since the Earth is approximately 70% covered by water, validation of remote sensing algorithms over water is particularly important. COVE provided over 16 years of continuous surface radiation and aerosol measurements at the Chesapeake Light Station for validation of CERES and other satellite products (MISR, MODIS, SeaWiFS etc.). Observations ended in 2016 due to structural concerns. To our knowledge, no other ocean platform provided the type of continuity or quality equal to COVE (Rutledge et al., 2006).

Upwelling irradiance measurements over water are a challenge because 1.) They require a fixed platform in order to maintain a precise downward viewing geometry, 2.) The instruments must be mounted high enough above the water's surface to avoid spray, and 3.) The fixed platform should not occupy a significant portion of the instrument field of view (FOV); this is especially difficult for upwelling measurements over water.

There are other over-water sites that collect upwelling measurements, but many sites focus on measuring water-leaving radiances (e.g., to infer chlorophyll-a) and therefore are not impacted by the structure that supports the instruments (e.g., Hooker et al., 2003; Zibordi et al., 2006, 2009; Ha et al., 2019). As far as we know, the only other upwelling irradiance measurements collected at a site similar to COVE was an experiment at Buzzards Bay Entrance Light Station off the coast of Massachusetts, where Payne (1972) mounted shortwave pyranometers on a short 2 m boom to measure albedo. Not surprisingly, Payne (1972) also had obstruction issues and deleted the contaminated afternoon data.

At COVE, upwelling longwave radiation  $(LW_{prg}^{\uparrow})$  was measured with an Eppley pyrgeometer (model PIR), which has a broadband spectral range of 4-50  $\mu$ m. Ideally, pyrgeometers should be installed at a height of 30 m and located in a position with an unobstructed hemispherical FOV (McArthur, 2005). That installation was impractical at the Chesapeake Lighthouse. Thus, we located the upwelling instrumentation on an existing 8 m boom, 21 m above the surface. However, the 8 m boom length was insufficient to prevent the main structure from contributing a significant radiation signature to the pyrgeometer FOV. That is, the west side of the structure (where the boom is located) blocked an estimated 15% of the pyrgeometer irradiance (see geometry in Figure 1 and Figure 2), which caused anomalies in the pyrgeometer measurements.

Fortunately, there were other measurements at COVE that can be used to determine  $LW^{\uparrow}$  without the upwelling hemispheric FOV pyrgeometer. Here, we describe how to use SST, downwelling longwave  $(LW^{\downarrow})$ , and temperature measurements in an energy balance equation to derive  $LW^{\uparrow}$  values that are not affected by radiation emitted from the lighthouse. This component summation technique allows us to recover 10+ years of  $LW^{\uparrow}$  data at COVE that adheres to the rigorous BSRN target uncertainties and provides the best possible  $LW^{\uparrow}$  measurements in the presence of an obstruction. We also discuss how our results can be applied to other locations with different measurement geometries and obstruction issues, and suggest that others use the  $LW^{\uparrow}$  component summation technique to verify their own  $LW^{\uparrow}$  measurements.

## 2 Motivation

75

The  $LW^{\uparrow}$  measurement problem at COVE is illustrated quantitatively in Fig. 3, which presents differences between wide FOV upwelling pyrgeometer data and narrow FOV IRT data as a function of the solar geometry for the year 2008. The datapoints

are very small so that daily data appear as U-shaped 'lines'. The highest solar zenith angles at solar noon (when solar azimuth equals 180°) occur during the winter months, so lines for the winter season appear at the top of the figure. Likewise, the summer season occupies the lowest lines. The black dashed line in the middle represents the equinoxes.

One can follow a U-shaped line to surmise how the instrument differences vary throughout a day; the green and blue lines at the top of the figure indicate rather steady differences throughout winter days, whereas the lines below the equinoxes indicate  $\sim 30 \text{ W-m}^{-2}$  to  $\sim 50 \text{ W-m}^{-2}$  of variability throughout a typical summer day. The large range of biases (almost always positive) shown in Fig. 3 indicates that the lighthouse structure effect can be quite significant.

Ideally, the differences between the two measurements should remain steady throughout the day. However, the lighthouse structure has a different effective temperature than the water that it is blocking from the pyrgeometer, and this temperature difference has diurnal and seasonal cycles. It is also notable that the diurnal variability of the differences is greatest in the summer months, with the darkest reds occurring in the afternoons at azimuths between  $\sim 200$  and  $300^{\circ}$ . This 'Summer Afternoon Anomaly' is caused by the structure heating up throughout the clear-sky days while the ocean temperature remains steady.

Although the IRT measurements are useful for demonstrating the effect that the lighthouse structure has on the wide-FOV pyrgeometer measurements, the IRT is not appropriate for monitoring upwelling longwave radiation. This is because the IRT measurements do not capture the reflected  $LW^{\downarrow}$  or the air emission between the water and the pyrgeometer. We consider the effects of these additional components in Section 4 and demonstrate that the seasonal and inter-annual variability of this "lighthouse structure effect" is often greater than the maximum 2% bias recommended by the BSRN.

## 3 Instrumentation

100

110

The instruments used in this study are the following:

- **Pyrgeometers**: The pyrgeometer measures longwave irradiance. A pyrgeometer is intended for unidirectional operation in the measurement of  $LW^{\downarrow}$  or  $LW^{\uparrow}$  irradiance. We used a Precision Infrared Radiometer (PIR) from Eppley Laboratories. The angular FOV is 180° (hemispherical FOV is  $2\pi$  sr). Spectral Range is approximately 4-50  $\mu$ m and uncertainty is 5 W-m<sup>-2</sup> (www.eppleylab.com/instrument-list/precision-infared-radiometer/). Calibrations were typically every 2-3 years.
- Infrared Radiation Thermometer (IRT): The IRT measures the ocean skin temperature. It is a Heitronics model KT19.85 with a spectral range of 9.6-11.5  $\mu$ m. This spectral range is where common atmospheric gases (including water vapor) do not absorb (i.e. atmospheric window). The lens used had an angular FOV of either 2.8° or 8.9°, depending on which instrument was in use during the calibration cycle. Accuracy is  $\pm$  0.5°C, plus 0.7% of the temperature difference between the housing containing the measuring instrument and the object to be measured (www.heitronics.com/wp-content/uploads/KT19.85-II-Datenblatt-EN-05.pdf). Calibrations were done typically every 1-2 years. The IRT was looking down at the water at an approximate 45° angle.
- **Meteorological**: Air temperature and Relative Humidity (RH) were measured with both a Vaisala (model CS500 and HMP50) and Rotronic (model HC-S3). Temperature accuracy is approximately  $\pm$  0.5°C for Vaisala and  $\pm$  0.3°C for Rotronic. The models listed above contain a Platinum Resistance Temperature detector to measure air temperature and require minimal

maintenance. The RH from both the CS500 and HMP50 used an intercap sensor with accuracy of 3% between 10-90% RH and 6% between 90-100% RH, with improved accuracy when switched to the Rotronic Hygroclip S3 sensor with accuracy of 1.5% (https://s.campbellsci.com/documents/us/manuals/cs500.pdf, https://s.campbellsci.com/documents/us/manuals/hmp50.pdf, and https://s.campbellsci.com/documents/ca/manuals/hc-s3\_man.pdf). Temperature calibration and/or replacement of probes were conducted approximately every 2-3 years. In place of calibrations, the RH chip would be replaced as needed. Atmospheric pressure measurements were made with a Vaisala sensor, model PTB101B. The accuracy was ±0.5 mb at 20°C and ±1.5 mb from 0-40°C (https://s.campbellsci.com/documents/eu-/brochures/Manuals/ptb101b.pdf). Calibrations were made every 1-2 years.

- Ground-Based Global Positioning System (GPS) Meteorology: Measures integrated (total column) precipitable water vapor (in cm) in the atmosphere. Determined from a Global Positioning System (GPS) receiver and measured in centimeters every 30 minutes. GPS satellite observations are combined with GPS satellite orbit and Earth orientation parameters to estimate GPS signal delay (Zenith Total Delay or ZTD). Signal delays are then combined with surface meteorological information to estimate total precipitable water (Holub and Gutman, 2016). Meteorological calibrations generally occurred every 1-2 years, the physical GPS receiver did not require calibration.

## 4 Method

125

130

Although the lighthouse structure is a necessary fixture at COVE, it is also an undesirable interference that reduces the accuracy of the flux measurements. That is, the pyrgeometer measurements  $(LW_{prg}^{\uparrow})$  at COVE had a portion of the FOV blocked by the Chesapeake Lighthouse (Figure 1 and Figure 2), so the measurements can be expressed as:

$$135 \quad LW_{prq}^{\uparrow} = (1-f)LW_{f=0}^{\uparrow} + fLW_{twr}^{\uparrow}, \tag{1}$$

where f is the fraction of the upwelling irradiance that is blocked by the structure,  $LW_{f=0}^{\uparrow}$  is the upwelling longwave flux in the absence of the structure, and  $LW_{twr}^{\uparrow}$  is the upwelling longwave flux emitted by the structure. The upwelling irradiance blockage can be computed analytically for simple structures, as shown in Appendix A.

Note that when f is small enough,  $LW_{prg}^{\uparrow} \simeq LW_{f=0}^{\uparrow}$  and the pyrgeometer measurements accurately represent the upwelling longwave radiation. However,  $f \simeq 0.15$  at COVE, so the pyrgeometer measurement does not accurately represent  $LW_{f=0}^{\uparrow}$ . The longwave component summation method described in Section 4.1 is a technique for obtaining  $LW_{f=0}^{\uparrow}$  without using a wide field-of-view pyrgeometer or determining f.

We assume that objects emit negligible radiation outside of the spectral range of the pyrgeometer at terrestrial temperatures throughout this paper. Thus, we have approximated the Stefan-Boltzmann law as

145 
$$\int_{A_{\rm HID}}^{50\,\mu\rm m} \varepsilon(\lambda)B(T,\lambda)d\lambda \simeq \int_{0}^{\infty} \varepsilon(\lambda)B(T,\lambda)d\lambda = \frac{\varepsilon_b \sigma T^4}{\pi},\tag{2}$$

where  $\varepsilon(\lambda)$  is the emissivity at wavelength  $\lambda$ ,  $B(T,\lambda)$  is the Planck function at temperature T and wavelength  $\lambda$ , and  $\varepsilon_b$  is the broadband emissivity of a medium. We demonstrate how to compute the broadband emissivity of water in Appendix B.

# 4.1 The Longwave Component Summation Method

165

The air temperature and skin temperature of the water within the small field of view of the down-looking measurements at COVE are very likely isothermal, so the upwelling radiation at COVE is isotropic (Stephens, 1994, Section 7.1). Since the air temperature and water skin temperature are horizontally homogeneous, we can assume that they are both gray bodies and we can use the Stefan-Boltzmann law to relate Planck's function to broadband irradiance. Hence, we can use an energy balance (e.g., Lee et al., 2010) to express  $LW_{f=0}^{\uparrow}$  as a component summation of measurements that do not require a pyrgeometer and are not influenced by the lighthouse:

155 
$$LW_{f=0}^{\uparrow} = LW_{cs}^{\uparrow} = (1 - \varepsilon_1)[\varepsilon_w \sigma T_w^4 + (1 - \varepsilon_1)(1 - \varepsilon_w)LW^{\downarrow}] + \varepsilon_1 \sigma T_1^4,$$
 (3)

where  $LW_{cs}^{\uparrow}$  denotes upwelling longwave irradiance determined by "component summation." Here,  $\varepsilon$  denotes the broadband emissivity of a medium, T is the temperature of a medium,  $\sigma$  is the Stefan-Boltzmann constant  $(5.67 \times 10^{-8} \text{ W-m}^{-2} \cdot K^{-4})$ , and  $LW^{\downarrow}$  denotes the atmospheric downwelling longwave radiation measured at the site. The subscript 1 refers to the layer of air below the boom height, which we call Layer 1 (see  $\varepsilon_1$  in Figure 1); the w subscript represents water. Water emissivity is  $\varepsilon_w = 0.92$  (see Appendix B).

Equation 3 includes water emission after attenuation by the air below our sensors  $[(1-\varepsilon_1)(\varepsilon_w\sigma T_w^4)]$ , water reflectance after two-way attenuation by the air below our sensors  $[(1-\varepsilon_1)^2(1-\varepsilon_w)LW^{\downarrow}]$ , and emission of the air below our sensors  $(\varepsilon_1\sigma T_1^4)$ . We will see later in this paper that water emission is the dominant term, since the longwave reflectance of water is only  $(1-\varepsilon_w)=0.08$  and the emissivity of layer one  $(\varepsilon_1)$  is even smaller. Thus, downwelling atmospheric radiation that is reflected off of the water is an order of magnitude less than the irradiance emitted by the water, and the irradiance of the air between the boom and the water is two orders of magnitude less than the irradiance of the water below.

We can also determine the LW upwelling immediately above the ocean surface by omitting Layer 1 from the component sum. This is easily accomplished by setting the emissivity of layer one to zero (i.e.,  $\varepsilon_1 = 0$ ) to obtain:

$$LW_{cs, srfc}^{\uparrow} = \varepsilon_w \sigma T_w^4 + (1 - \varepsilon_w) LW^{\downarrow}. \tag{4}$$

The surface upwelling is more useful for satellite comparisons than measurements above Layer 1, so we include this value in the data archives as well.

Further discussion about the origin and typical magnitudes of each term in Equation 3 are available in Appendix C. However, we need to determine  $\varepsilon_1$  before we can compute  $LW_{cs}^{\uparrow}$  at the boom height.

## 4.2 Determining the Emissivity of the Air Below the Upwelling Instruments (Layer 1)

Calculating  $LW_{cs}^{\uparrow}$  with Equation 3 is straightforward using measurements at COVE, except for the emissivity of Layer 1 ( $\varepsilon_1$ ). However, the emissivity of the atmospheric column ( $\varepsilon_{atm}$ ) ranges from  $\sim 0.6$ -0.9 (Zhao et al., 2019, and discussion in Appendix D) and is relatable to  $\varepsilon_1$ . Additionally, we note that nearly all of the longwave absorption in the atmosphere is

attributable to water vapor; hence, we determine the relationship between  $\varepsilon_1$  and  $\varepsilon_{atm}$  by scaling the ratio of longwave optical depths (Layer 1 / atmospheric column) with the corresponding ratio of water vapor. That is, the scale factor  $\eta$  for the longwave optical depth can be expressed as:

$$\eta = \frac{\tau_1}{\tau_{atm}} = \left\lceil \frac{Q_1 \rho_1 Z_1}{W} \right\rceil. \tag{5}$$

Here,  $\tau_1$  and  $\tau_{atm}$  denote the longwave optical depths corresponding to Layer 1 and the atmospheric column, and the term in the brackets is the ratio of the water vapor in Layer 1 to the column precipitable water vapor, a methodology resembling (Liu, 1986). The variables in the square brackets are determined by using four years of measurements (2004-2007) at COVE when precipitable water vapor was available:  $Q_1$  is the water vapor mixing ratio obtained from temperature and pressure data (Wallace and Hobbs, 2006, page 82),  $\rho_1$  is the density of dry air at standard temperature and pressure (1.225 kg/m³),  $Z_1$  is the boom height (21m), and W is the column precipitable water vapor obtained from GPS-Met (Holub and Gutman, 2016).

Recalling that transmissivity  $(e^{-\tau})$  is related to emissivity by  $e^{-\tau} = 1 - \varepsilon$ , Equation 5 can also be expressed as:

$$\eta = \frac{\ln(1 - \varepsilon_1)}{\ln(1 - \varepsilon_{atm})} = \left\lceil \frac{Q_1 \rho_1 Z_1}{W} \right\rceil,\tag{6}$$

90 where  $\varepsilon_{atm}$  is the emissivity of the atmospheric column. Solving for  $\varepsilon_1$ , we obtain:

$$\varepsilon_1 = 1 - (1 - \varepsilon_{atm})^{\eta}. \tag{7}$$

Thus, we can compute the emissivity of the air below our sensors using the emissivity of the atmosphere and meteorological data obtained at the site. We do not have instrumentation for obtaining  $\varepsilon_{atm}$  (or  $\tau_{atm}$ ), but documented clear-sky atmospheric emissivities range from  $\varepsilon_{atm} \simeq 0.6$  to 0.9 (e.g., Zhao et al., 2019), so we choose a midrange value of  $\varepsilon_{atm} = 0.75$ . We discuss the ramifications of this assumption in Appendix D. We used four years of meteorological data (2004-2007) with  $\varepsilon_{atm} = 0.75$  to obtain a median  $\varepsilon_1 = 0.015$  and inter-quartile range = 0.007. We also note that the water vapor scale factor  $\eta$  at COVE has a median value of 0.011 over the same time period, with the 25-75 percentile ranging from 0.009 to 0.014. Liu (1986) found that  $\eta$  is relatively constant at  $\sim$ 46 mid-ocean and small islands stations when  $W \lesssim 4$  g-cm<sup>-2</sup>, so it is not surprising that we found very little difference between using variable  $\eta$  obtained from meteorological data and using a climatological median for  $\eta$ ; this is discussed further in Appendix D.

## 4.3 Longwave Component Summation Uncertainty

We used a statistical simulation to quantify the uncertainty in the longwave component summation. We started by using the annual mean climatology at the COVE site to represent the baseline condition:  $T_1 = 289$  K for air temperature,  $T_w = 290$  K for water temperature,  $\varepsilon_1 = 0.015$  for the emissivity of the air in Layer 1,  $\varepsilon_w = 0.92$  for the emissivity of water, and  $LW^{\downarrow} = 339$  W-m<sup>-2</sup> for the downwelling longwave irradiance. Then we assumed Gaussian distributions for the variables in Equation

3, using standard deviations of  $\delta T_1 = \pm 0.5$  K, and  $\delta T_w = \pm 0.5$  K,  $\delta \varepsilon_1 = \pm 0.007$ ,  $\delta \varepsilon_w = \pm 0.001$ , and  $\delta LW^{\downarrow} = \pm 5$  W-m<sup>-2</sup>. We simulated one million  $LW^{\uparrow}_{cs}$  computations using these ranges of values and analyzed the differences with respect to the baseline condition.

Specifically, we computed the independent errors associated with random Gaussian noise for five variables. These error values were then added to their baseline values to form the basis of one million simulated observations, and the  $LW_{cs}^{\uparrow}$  values for all cases were then calculated using Equation 3. We then computed the differences between these 'perturbed'  $LW_{cs}^{\uparrow}$  values and the baseline  $LW_{cs}^{\uparrow}$  and analyzed the results. The simulated mean bias in  $LW_{cs}^{\uparrow}$  using these assumptions is less than 0.01 W-m<sup>-2</sup> and the standard error is 2.5 W-m<sup>-2</sup>. The relative standard error in  $LW_{cs}^{\uparrow}$  is 0.6% (i.e., standard error divided by baseline  $LW_{cs}^{\uparrow}$ ), which is well within the BSRN target uncertainty of 2%.

#### 215 5 Results

The evaluation of  $\varepsilon_1$  in Section 4.2 allows us to use Equation 3 to quantify the upwelling LW component summation  $(LW_{cs}^{\uparrow})$  during the 10 years of IRT measurements at COVE; this data is now publicly available at the NASA LaRC OpenData archive and has been submitted to the BSRN database. In this section, we argue that the upwelling LW component summation  $(LW_{cs}^{\uparrow})$  is more accurate then upwelling pyrgeometer measurements  $(LW_{prg}^{\uparrow})$  at COVE or any other location that has a significant obstruction in the field of view.

We begin by showing that the  $LW_{cs}^{\uparrow} - LW_{IRT}^{\uparrow}$  differential does not exhibit the same Summer Afternoon Anomaly that we presented in Section 2 (Figure 3) for the pyrgeometer measurements. Next, we present four single-day scenarios that illustrate how solar heating of the lighthouse and changes in air temperature perturb the pyrgeometer measurements. Finally, we present the monthly and annual pyrgeometer biases with respect to the component summation technique, noting that the pyrgeometer biases are often greater than the 2% BSRN target uncertainty.

## 5.1 The Disappearance of the Summer Afternoon Anomaly

Recall the Summer Afternoon Anomaly of Figure 3, where  $LW_{prg}^{\uparrow} - LW_{IRT}^{\uparrow}$  produced anomalously high values that are associated with solar heating of the lighthouse on sunny summer days. In this section we demonstrate that a similar anomoly does not occur for  $LW_{cs}^{\uparrow} - LW_{IRT}^{\uparrow}$  when we use component summations for the upwelling irradiance.

Noting that  $LW_{IRT}^{\uparrow} = \varepsilon_w \sigma T_w^4$ , we solve Equation 3 for the difference between the component summation irradiance and the IRT measurements:

$$LW_{cs}^{\uparrow} - LW_{LRT}^{\uparrow} = -\varepsilon_1 \varepsilon_w \sigma T_w^4 + (1 - \varepsilon_1)^2 (1 - \varepsilon_w) LW^{\downarrow} + \varepsilon_1 \sigma T_1^4. \tag{8}$$

The significance of the various terms in Equation 8 are most easily explained by inserting  $\varepsilon_1 = 0.015$  and  $\varepsilon_w = 0.92$  (from Section 4.2 and Appendix B) into Equation 8:

$$LW_{cs}^{\uparrow} - LW_{LRT}^{\uparrow} = 0.015 \times (\sigma T_1^4 - 0.92 \times \sigma T_w^4) + 0.078 \times LW^{\downarrow}.$$
 (9)

Notice that the right hand side of Equation 9 is dominated by the reflected  $LW^{\downarrow}$ , since the coefficients for the atmosphere and water emission terms are a factor of 5 smaller than the coefficient for the longwave reflectance (i.e.,  $0.078/0.015 \simeq 5$ ). The air and water emission terms also tend to cancel one another, since  $T_1 \simeq T_w$  on the Kelvin temperature scale. Taken together, they contribute less than 1 W-m<sup>-2</sup> to Equation 9 when there is a 10 K temperature difference between the air and the water. Meanwhile, values for  $LW^{\downarrow}$  at COVE are about 170 – 470 W-m<sup>-2</sup>, so the 3rd term in Equation 9 contributes 13 – 37 W-m<sup>-2</sup>. Thus,  $LW^{\uparrow}_{CS} - LW^{\uparrow}_{LRT} \simeq (0.078)LW^{\downarrow}$  when  $\varepsilon_w = 0.92$ .

The diurnal variability of Equation 8 is shown in Figure 4. Note that the summer afternoon anomaly (or 'hotspot') shown by the red points in Figure 3 does not occur in Figure 4 and that the pattern is much more symmetric after solar noon (i.e., symmetric after solar azimuth angle of 180 degrees). The biases still vary significantly between the summer and winter months, but this is because the reflected downwelling longwave  $(0.078 \times LW^{\downarrow})$  is greater in the summer than in the winter.

# 5.2 Single Day Illustrations of Pyrgeometer Anomalies

In this section we analyze four single-day scenarios in winter and summer and in clear and overcast conditions to help understand the physics driving the biases in Figures 3 and 4. Since the water emission term in Equation 3 ( $\varepsilon_w \sigma T^4$ ) dominates the radiative flux (see Appendix C), we expect the true upwelling flux to track the IRT measurements throughout the day (albeit with a high bias) and only minor perturbations associated with the air temperature. In the following paragraphs, though, we see that the pyrgeometer measurements can be significantly affected by changes in air temperature and/or solar heating of the lighthouse structure.

Single-day continuous measurements are presented in Figures 5 and 6. Conventions used in both figures are as follows:

- Solid black lines represent the  $LW_{prq}^{\uparrow}$  measurements and therefore include the lighthouse in the field of view.
- The light blue line is the water emission derived from the IRT  $(LW_{IRT}^{\uparrow})$  and therefore does not include emissions from the lighthouse or the air above the water.
  - The white line is derived from Equation 3 using  $\varepsilon_1 = 0.015$ .
  - The red line is air temperature and red squares are water temperatures at selected times of the day (right Y-axis).
  - Finally, the yellow shaded region denotes the solar elevation on that day (no scale).
- The bias in  $LW_{prg}^{\uparrow}$  caused by the lighthouse structure is clearly observed in the two winter scenarios of Figure 5. That is,  $LW_{prg}^{\uparrow}$  (the black line) are noticeably lower than  $LW_{cs}^{\uparrow}$  (white line) in Figure 5 throughout the day. This occurs when the air temperature is significantly colder than sea surface temperature since the effective temperature of the lighthouse structure above water is likely close to the air temperature, the cold structure obscures some of the warm water in the pyrgeometer field of view and lowers the irradiance at the instrument location. It is also apparent that  $LW_{prg}^{\uparrow}$  trends strongly with air temperature for the overcast winter day in the right panel of Figure 5 (comparing the red and black lines), further indicating that the structure temperature is tracking with the air temperature and affecting the pyrgeometer measurements. Meanwhile,  $LW_{cs}^{\uparrow}$  is

largely dominated by  $LW_{IRT}^{\uparrow}$  and does not include any terms that are affected by the light station. Hence, the white line tracks  $LW_{IRT}^{\uparrow}$  and are minimally affected by changes in the air temperature.

The differences between  $LW_{cs}^{\uparrow}$  (white line) and the  $LW_{prg}^{\uparrow}$  measurement (black line) is much greater on the clear winter day than the overcast winter day of Figure 5. This is because the temperature differential between the air (ergo, the lighthouse) and water is greater on the clear day than the overcast day. Nonetheless, we still see  $LW_{prg}^{\uparrow}$  rapidly responding to air temperature changes in Figure 5 on the overcast day as a response to changes in the lighthouse structure temperature. Note that the differences between  $LW_{cs}^{\uparrow}$  and the  $LW_{prg}^{\uparrow}$  measurement can be quite high on these winter days (up to  $\sim 12 \text{ W-m}^{-2}$  or 3.2%) which indicates that  $LW_{prg}^{\uparrow}$  is outside the recommended BSRN target uncertainty of 2%. Finally, note that  $LW_{prg}^{\uparrow}$  is always less than  $LW_{cs}^{\uparrow}$  on both clear and overcast winter days because the lighthouse is colder than the water when the air is colder than the water.

The lighthouse structure effect on the  $LW^{\uparrow}_{prg}$  measurements is also distinct on the clear summer day shown in the left panel of Figure 6 (black line). Beginning with the pre-dawn portion of the day (i.e., solar azimuths less than  $\sim 60^{\circ}$ ), we see that  $LW^{\uparrow}_{prg}$  is slightly greater than  $LW^{\uparrow}_{cs}$ . As the sun rises, the air temperature (red line), water temperature (red squares) and the  $LW^{\uparrow}_{IRT}$  (light blue line) respond to the increasing insolation. The pyrgeometer (black line) responds dramatically to the heating of the lighthouse on this clear day when the structure is effectively warmer than the surrounding air in these conditions. Later in the day, the pyrgeometer responds to the decreasing insolation associated with lower solar elevation and returns to a value that is consistent with the previous night's value.

We see a different story on a summer overcast day shown in the right panel of Figure 6. Here, the nighttime air temperature before sunrise is about 1 K warmer than the water temperature, so the black line (pyrgeometer) is elevated above  $LW_{cs}^{\uparrow}$  (white line). As the day progresses, a blast of warm air after sunset heats the lighthouse and the pyrgeometer responds. Here again,  $LW_{cs}^{\uparrow}$  tracks the IRT measurements and does not directly respond to changes in air temperature. Thus, the component summation LW radiation ( $LW_{cs}^{\uparrow}$ , white line) tracks  $LW_{IRT}^{\uparrow}$  (light blue line) on all four days in Figures 5 and 6 and is not sensitive to air temperature. Notably, the air temperature in the shallow layer beneath the boom (layer 1) minimally affects  $LW_{cs}^{\uparrow}$ , with further details provided in Appendix C.

Finally, note that there is a period of time on the summer overcast day when the pyrgeometer provides the correct irradiance (from about solar noon until sunset in the right panel of Figure 6). This occurs when the water, air, and structure temperatures are in equilibrium (since the structure is at the same temperature as the water that it is blocking from the FOV of the pyrgeometer). It is also notable that the air and water temperatures also achieve equilibrium in the middle of the clear summer day (solar azimuth of  $210^{\circ}$ ), but the pyrgeometer is biased about 14 W-m<sup>-2</sup> high of  $LW_{cs}^{\uparrow}$ ; this is because of the direct sun heating the lighthouse above the ambient water temperature.

In summary, the monthly LW flux biases in Figures 7 and the four daily scenarios of Figures 5 & 6 indicate that:

1. Large and negative pyrgeometer biases (with respect to the component summation LW flux) occur in the winter because the lighthouse structure and air temperature are much colder than the water temperature this time of year.

- 2. Likewise, positive biases can occur on summer days when the lighthouse structure and air temperature are greater than the water temperature, especially on clear days when insolation directly warms the lighthouse structure.
- 3. The pyrgeometer provides the correct irradiance when the air temperature is in equilibrium with the water temperature in overcast conditions, which can occur any time of the year (per the range of values in the boxplots of Figure 7). However, this is not necessarily the case in clear-sky conditions.

## 305 5.3 Monthly and Annual Variations of Pyrgeometer Biases

Pyrgeometer biases with respect to the component summation method are related to the air-water temperature differential and to cloud conditions, so the biases are not constant. The monthly and yearly relative biases of the pyrgeometer with respect to the component summation  $LW_{cs}^{\uparrow}$  are summarized with boxplots in Figure 7. Boxes in the boxplots throughout this article indicate the interquartile range (IQR) and contain 50% of the data, the whiskers capture the 99 percentile, and the medians are denoted by red lines in the center of the boxes. All the boxplots have notches, which can barely be seen in Figure 7. The top and bottom edges of the notched regions correspond to median $+(1.57 \cdot IQR)/\sqrt{n}$  and median  $-(1.57 \cdot IQR)/\sqrt{n}$ . One can conclude with 95% confidence that the medians of two boxplots are different when the notches of the boxplots do not overlap (McGill et al., 1978). Lastly, the numbers at the top indicate the percentage of data outside the BSRN target uncertainty of  $\pm 2\%$  for each month, year and overall.

The left panel of Figure 7 presents the monthly variability of the  $LW_{prg}^{\uparrow}$  bias relative to  $LW_{cs}^{\uparrow}$  for 10 years of data (2004-2013). The absolute relative bias is largest in the coldest months and smallest in the warmest months. This is because the air is much colder than the water in the winter and winter skies tend to be cloudier than summer skies at the Chesapeake Lighthouse, so the lighthouse is generally colder than the water in the winter. Since the cold lighthouse is blocking some of the warm water from the field of view of the pyrgeometer, the pyrgeometer registers less upwelling flux than it would if water filled the entire field of view. Likewise, the air and water temperatures have similar values in the summer months, so the tower temperature is close to the water temperature and the pyrgeometer bias tends to be smaller than in the winter months (for similar cloud conditions). Additionally, solar heating of the tower is greater and the days are longer in the summer months than in the winter months, and this pushes the median pyrgeometer bias from negative to positive during the warmest months of the year.

Although the median pyrgeometer bias is within the 2% BSRN target uncertainty for every month, more than 25% of the data has biases greater than  $\sim$ 2% for the winter months (Dec, Jan, and Feb) and one spring month (Mar). Additionally, the whiskers (lower and upper) indicate that 8 months have at least 15% of the data outside the target uncertainty; thus, a substantial portion of the pyrgeometer data do not conform to the BSRN target uncertainty. The amount of data outside the 2% target uncertainty ranges from 6.0% in August to 35.2% in January.

The right panels of Figure 7 show the inter-annual variability. Here again, the whiskers indicate that the pyrgeometer bias with respect to the component summation irradiance is greater than  $\pm 2\%$  for a substantial portion of the data. The amount of annual data outside the target uncertainty range from a low of 10.8% in 2006 to a high of 26.9% in 2007. The single box and whisker plot at the end of Figure 7 is the result of the entire 10 year period and displays  $\sim 18.2\%$  of the data is outside the

target uncertainty. This illustrates a large amount of data is outside the BSRN target uncertainty, which we attribute this to the lighthouse structure influencing the  $LW_{prg}^{\uparrow}$  measurements.

Note that years 2005 and 2009-2011 had negative median pyrgeometer biases that correspond to the strongest negative airwater biases (shown in the bottom of the figure). Recall that negative biases are expected when the lighthouse is in equilibrium with the air and the air is colder than the water (on a median basis, the air is always colder than the water at the Chesapeake Lighthouse), so it is no surprise that strong negative air-water temperature biases result in a negative median pyrgeometer bias. However, solar heating of the lighthouse can result in effective lighthouse temperatures that are greater than both the surrounding air and the water; when this occurs,  $LW_{prg}^{\uparrow} > LW_{cs}^{\uparrow}$  and the pyrgeometer bias is positive. Thus, cold cloudy winters tend to favor negative annual pyrgeometer biases, whereas hot sunny summers favor positive annual pyrgeometer biases.

## 6 Discussion

# 6.1 Alternate Geometries with Different Obstruction Issues

Thus far, we have presented results specific to the geometry of the COVE platform, which obstructs about 15% of the pyrgeometer upwelling measurement (i.e.,  $f \simeq 0.15$ ). The obstruction percentage was calculated using a software package called ImageJ (https://imagej.net/ij/), which provides area and pixel value calculations within manually selected regions. In our application, we used ImageJ to distinguish the structure from the water surface and to estimate the percentage of the upwelling occupied by the structure. We conservatively estimate the accuracy of f as  $\pm 0.05$  because the camera used for Figure 2 is not precisely positioned at the pyrgeometer location and it may not be exactly level. Additional discussion about how f is affected by an obstruction's geometry is provided in Appendix A.

In this section we use the measurements at COVE to compute how measurements would be impacted for platform geometries that are different than the Chesapeake Lighthouse (i.e.,  $f \neq 0.15$ ). First, we insert f = 0.15 into Equation 1 to describe the  $LW^{\uparrow}$  measured with the pyrgeometer when an obstruction blocks 15% of the upwelling irradiance:

$$LW_{p15}^{\uparrow} \equiv LW_{prg}^{\uparrow}(f=0.15) = 0.85 \times LW_{f=0}^{\uparrow} + 0.15 \times LW_{twr}^{\uparrow},$$
 (10)

Solving both Equation 1 and Equation 10 for  $LW_{twr}^{\uparrow}$  and equating the resulting expressions, we obtain an expression for the perturbation  $LW_{prq'}^{\uparrow} - LW_{cs}^{\uparrow}$  associated with any value of f:

$$LW_{prg'}^{\uparrow} - LW_{cs}^{\uparrow} = \left[ \frac{LW_{p15}^{\uparrow} - LW_{cs}^{\uparrow}}{0.15} \right] f, \tag{11}$$

where the prg' subscript indicates that the pyrgeometer is not fixed at a location where f=0.15, and f can have any value between 0 and 1. COVE pyrgeometer measurements provide  $LW_{p15}^{\uparrow}$ , and  $LW_{cs}^{\uparrow}$  is obtained from Equation 3 using the IRT,  $LW^{\downarrow}$ , and ambient air temperature  $(T_1)$  measurements at COVE. Dividing Equation 11 by  $LW_{cs}^{\uparrow}$  yields the relative bias

associated with the lighthouse structure perturbation:

$$\frac{LW_{prg'}^{\uparrow} - LW_{cs}^{\uparrow}}{LW_{cs}^{\uparrow}} = \left[\frac{LW_{p15}^{\uparrow} - LW_{cs}^{\uparrow}}{(0.15)LW_{cs}^{\uparrow}}\right]f. \tag{12}$$

which is shown for discrete values of f in Figure 8.

385

390

The box and whiskers at f=0.15 in Figure 8 correspond to measurements at the COVE site, and is identical to the climatological box and whiskers for Years 2004-2013 in the rightmost panel of Figure 7. The remaining box and whiskers in Figure 8 are obtained by computing the relative bias using variable f in Equation 12. As discussed earlier, more than 18% of the  $LW_{prg}^{\uparrow}$  data at the COVE site has biases greater than the BSRN requirement of 2% (with respect to  $LW_{cs}$ ). However, Figure 8 also indicates that the same dataset would produce biases of less than 2% for nearly all of the data if the pyrgeometer would have been located far enough away from the lighthouse such that  $f \lesssim 5\%$ . Unfortunately, the Chesapeake Lighthouse is so large that the pyrgeometer would need to be located on a 14m boom (an additional 6 m longer than the 8 m boom that is part of the Lighthouse) to achieve this level of agreement between  $LW_{prg}^{\uparrow}$  and  $LW_{cs}^{\uparrow}$ . However, instruments located on platforms smaller than the Chesapeake Lighthouse should easily achieve f < 5%, but this should always be verified before establishing new sites.

# 6.2 The Longwave Component Summation as a Residual Check for Longwave Flux Measurements

The BSRN protocols require some measurement redundancy in order to assure accuracy and mitigate data loss. For example, it is standard procedure for site managers to report shortwave irradiance using two different measurement techniques (Ohmura et al., 1998; McArthur, 2005; Driemel et al., 2018). These measurement techniques are 1.) The shortwave component summation method, which sums direct normal irradiance measurements and shaded diffuse irradiance measurements, and 2.) the global method, which utilizes an unshaded pyranometer. The residual differences between these two methods are an important tool for verifying that the solar tracker is working properly.

In this section, we propose using Equation 3 as a component summation method for  $LW^{\uparrow}$ . The basic premise is that all BSRN sites have  $LW^{\downarrow}$  and ambient air temperature measurements, so the only additional instrumentation needed to compute  $LW^{\uparrow}$  with Equation 3 is a precision IRT for obtaining  $T_w$  (or  $T_l$  over land). This redundancy could potentially discover a drifting pyrgeometer long before the instrument was due for calibration.

Additionally, this  $LW^{\uparrow}$  component summation approach could verify the quality of the pyrgeometer measurements at a site. For instance, Figure 7 demonstrates that the pyrgeometer measurements at the COVE site frequently do not meet the BSRN target accuracy (because the instrument is located too close to the structure). On the other hand, Figure 8 demonstrates that BSRN target accuracies could be achieved for 99% of the data for geometries where the structure obstruction occupies less than 5% of the field of view. Thus, the  $LW^{\uparrow}$  component summation technique can verify whether the structure obstruction is problematic for any BSRN site.

One potential drawback to the  $LW^{\uparrow}$  component summation technique is that the air and surface emissivities in Equation 3 are often unknown. The emissivity of the air below the instrumentation ( $\varepsilon_1$ ) can be derived from Equations 5-7 using standard meteorological data and the column precipitable water vapor. However,  $\varepsilon_1$  is small and the air below our pyrgeometer height of 21 m has very little effect on the irradiance. If column precipitable water vapor is not available, one can characterize the

site with temporary measurements to determine a single characteristic  $\varepsilon_1$  for that location. In our case, the impact of using a characteristic  $\varepsilon_1=0.015$  instead of computing near instantaneous  $\varepsilon_1$  had the highest maximum effect of  $\sim \pm 0.4$  W-m<sup>-2</sup> in the winter months and lowest maximum effect of  $\sim \pm 0.2$  W-m<sup>-2</sup> for spring (see Figure D1).

The other emissivity of concern in Equation 3 is  $\varepsilon_w$ , which needs to be replaced by its land-based cousin  $\varepsilon_l$  for land sites. Although water emissivity is characterized by  $\varepsilon_w=0.92$ , land surface emissivity depends upon the surface type (sand, silt, clay, vegetation, snow, cement, etc.) and land surface moisture. Values range from  $\sim 0.85-0.97$  for various land surface in the thermal infrared (Tian et al., 2019; Huang et al., 2016; Li et al., 2011), but surface emissivities could be lower in desert areas (https://www.jpl.nasa.gov). Site managers at non-water locations need to evaluate the surface emissivity at their sites and assess how the accuracy of their evaluation affects  $LW_{cs}^{\uparrow}$  before adopting the LW component summation method.

While our newly developed component summation approach provides an alternative method for obtaining  $LW^{\uparrow}$ , pyrgeometers offer operational simplicity via a single direct irradiance measurement (instead of using a combination of measurements) and remain the preferred option in unobstructed environments. This is because pyrgeometers account for reflected  $LW^{\downarrow}$  without requiring knowledge of the broadband emissivity for the surface or the air below the sensors. This is especially important over land, where surface emissivity can vary with seasonal vegetation changes, surface moisture, and drought. Hence, measurement networks like BSRN recommend pyrgeometers for longwave irradiance measurements, but note that BSRN will accept  $LW^{\uparrow}_{cs}$  as an ancillary approach.

## 7 Conclusions

We have described a longwave component summation method to measure upwelling longwave irradiance that does not include contributions from a host structure that supports the instruments. The technique requires a precision infrared thermometer, a  $LW^{\downarrow}$  measurement (for water reflectance), meteorological data, and perhaps column precipitable water vapor measurements (to aid in characterizing the emissivity of the air below the measurements, depending upon location). We also present four case studies in different conditions (i.e., winter and summer, clear and overcast) and discuss the contributions of the radiative components term-by-term (i.e., water emission, longwave reflectance, and the air below the instruments) so that the reader can gain perspective about the relative importance of each of the terms.

The longwave component summation technique indicates that  $LW_{prg}^{\uparrow}$  measurements display biases reaching 35% in January as the highest monthly bias, a 27% bias in 2007 as the highest bias in the yearly variability, and an 18% bias overall for the 10-year period. These biases are primarily driven by significant winter temperature differences between the water and air (and consequently the lighthouse structure). Conversely, summer solar heating can elevate lighthouse temperatures above both air and water temperatures, causing  $LW_{prg}^{\uparrow}$  to exceed  $LW_{cs}^{\uparrow}$  and generating positive pyrgeometer bias.

The Chesapeake Lighthouse uniquely occupies  $\sim 15\%$  of the field of view of the upwelling instruments. Thus, we used COVE data to estimate the "lighthouse structure effect" that might be observed at different locations with different geometries. We found that geometries where the support structure occupies 5% or less of the instrument field of view will have biases of less than  $\pm 2\%$  for 99% of the data. However, we recommend the longwave component summation technique as the primary

upwelling longwave irradiance method for all sites that have significant obstructions (such as on ships or large fixed structures over water).

We also provide some discussion about applying the longwave component summation technique to land sites. The main challenge associated with applying the longwave component summation method to land sites is to accurately assess the surface emissivity. Nonetheless, we propose that precision infrared thermometers be added to these surface radiation measurement sites as well, especially if land sites already collect  $LW^{\downarrow}$  and air temperature measurements, since the longwave component summation method can be used as a secondary method to monitor possible drift that may occur between calibrations of the primary irradiance measurements.

Finally, we evaluated  $LW_{cs}^{\uparrow}$  uncertainties through a statistical simulation. We determined a mean bias of less than 0.01 W-m<sup>-2</sup> and a standard error of 2.5 W-m<sup>-2</sup> by comparing perturbed  $LW_{cs}^{\uparrow}$  to baseline values. The resulting  $LW_{cs}^{\uparrow}$  relative standard error is approximately 0.6%, well within the BSRN target uncertainty criteria. Thus, it is important to use the longwave component summation technique whenever a host structure occupies a significant portion of the instrument field of view.

#### 440 Data availability.

All data is publicly available at https://science-data.larc.nasa.gov/LaRC-SD-Publications/2025-07-25-001-BEF/data/ and has been submitted to the BSRN archive at https://doi.pangaea.de/10.1594/PANGAEA.984135

# **Appendix A: Example Geometry**

The longwave upwelling flux from an isothermal surface can be expressed as (e.g. Bohren and Clothiaux, 2006, Section 4.2):

$$LW^{\uparrow} = L(T_s) \int_{0}^{2\pi} \int_{0}^{\pi/2} \cos \vartheta \sin \vartheta \, d\vartheta \, d\phi = \pi L(T_s),$$
 (A1)

where  $\vartheta$  is the viewing zenith angle,  $\phi$  is the azimuth angle, and  $L(T_s)$  is the radiance of the surface at temperature  $T_s$ . Noting that

$$\int \cos \vartheta \sin \vartheta \, d\vartheta = -\frac{1}{2} \cos^2 \vartheta,\tag{A2}$$

we can solve the integrals in Equation A1 to obtain

$$450 \quad LW^{\uparrow} = \pi L(T_s). \tag{A3}$$

When an obstruction is present, though, some of the surface is blocked. For example, if the rectangular cuboid shown in Figure A1 has a temperature of  $T_o$ , the upwelling flux is

$$LW^{\uparrow} = \pi L(T_s) - \left[ L(T_s) - L(T_o) \right] \int_{\delta\phi} \int_{\vartheta_{crit}}^{\pi/2} \cos\vartheta \sin\vartheta \, d\vartheta d\phi, \tag{A4}$$

where the critical zenith angle where the obstruction intersects the surface is

$$\vartheta_{crit} = \arctan(\frac{b}{h})$$
 (A5)

and total obstructed azimuth angle is

$$\delta\phi = \arctan(\frac{l_1}{b}) + \arctan(\frac{l_2}{b}). \tag{A6}$$

We solve the integral in Equation A4 with Equation A2 and multiply the last term in Equation A4 by  $(\frac{\pi}{\pi})$  to obtain

$$LW^{\uparrow} = \pi L(T_s) - \left[ L(T_s) - L(T_o) \right] \pi \frac{\delta \phi \cos^2 \vartheta_{crit}}{2\pi}.$$
 (A7)

Rearranging, this becomes:

$$LW^{\uparrow} = \left[1 - \frac{\delta\phi}{2\pi}\cos^2\vartheta_{crit}\right]\pi L(T_s) + \left[\frac{\delta\phi}{2\pi}\cos^2\vartheta_{crit}\right]\pi L(T_o) \tag{A8}$$

Recall that  $\pi L(T_s)$  is the upwelling flux from the surface in the absence of an obstruction (e.g.,  $LW_{f=0}^{\uparrow}$  in Equation 1),  $\pi L(T_o)$  is the upwelling from the obstruction (e.g.,  $LW_{twr}^{\uparrow}$  in Equation 1), and we see that the fraction of the upwelling flux that is blocked by the obstruction in Equation 1 is

$$465 \quad f = \frac{\delta \phi}{2\pi} \cos^2 \vartheta_{crit}. \tag{A9}$$

Figure A1. Simplified geometry for a pyrgeometer attached to a rectangular cuboid.

Note that f varies with  $\delta \phi$  and  $\vartheta_{crit}$ , but f is not sensitive to the size of the obstruction. Thus, we can decrease f by increasing  $\vartheta_{crit}$  (via a lower boom height h or a longer boom length b in Figure A1) or by utilizing narrow structures with small  $\delta \phi$ .

It is useful to look at two example cases. First, if a downlooking pyrgeometer is located high above the surface and close to a large building or the side of a ship, then  $b \ll h$ ,  $b \ll l_1$ , and  $b \ll l_2$  in Figure A1; thus  $\vartheta_{crit} \sim 0$  and  $\delta \phi \sim \pi$ . Computing f via Equation A9 in this case yields f=0.5; this means that half the surface flux is blocked by the obstruction when pyrgeometers are located on a short boom attached to a large structure.

Now, suppose we are lucky enough to locate our pyrgeometer on an 8 m boom that is 10 m above the water and centered on the bow of a ship with a 16 m beam (e.g., the Ron Brown; https://oceanexplorer.noaa.gov/technology/vessels/ronbrown/ronbrown.html). Then  $b=l_1$  (i.e., 16/2) =  $l_2 \Rightarrow \delta \phi = \pi/2$ , and  $\vartheta_{crit} = \arctan \frac{8}{10} = 39^\circ$ . In this case, Equation A9 indicates  $f = \frac{\cos^2(39)}{4} = 0.15$ , and 15% of the upwelling flux is perturbed by the obstruction. This still has a significant impact on the pyrgeometer irradiance measurements — as demonstrated in Section 5.2 — and the upwelling pyrgeometer measurements should be complemented or replaced by the component-sum technique.

The geometry of the Chesapeake Lighthouse is more complicated than a rectangular cuboid, so we used imaging software to obtain  $f \simeq 0.15 \pm 0.05$  for the lighthouse. Note that f is not needed for the component summation technique (Section 4.1, Equation 3). However, f is helpful for estimating the radiation perturbation caused by obstructions at other sites (Section 6.1).

# Appendix B: Broadband Emissivity of Water

The broadband emissivity of water  $(\varepsilon_w)$  can be computed by weighting the spectral emissivity of water  $(\tilde{\varepsilon}_{wtr}(\tilde{\nu}))$  with the Planck function over a defined wavelength range:

$$\varepsilon_{w} = \frac{\int_{\tilde{\nu}_{min}}^{\tilde{\nu}_{max}} \tilde{\varepsilon}_{wtr}(\tilde{\nu}) B_{\tilde{\nu}} \tilde{\nu}}{\int_{\tilde{\nu}_{min}}^{\tilde{\nu}_{max}} B_{\tilde{\nu}} d\tilde{\nu}}, \tag{B1}$$

where  $\tilde{\nu}$  is the wavenumber. The upwelling pyrgeometer measurement covers the 4-50  $\mu$ m range, so  $\tilde{\nu}_{min} = 200 \text{ cm}^{-1}$  and  $\tilde{\nu}_{max} = 2500 \text{ cm}^{-1}$  for our measurements. The wavenumber form of the Planck function (e.g., Stephens, 1994) is:

$$B_{\tilde{\nu}}(\tilde{\nu},T) = 2hc^2\tilde{\nu}^3 \left(\exp\left(\frac{hc\tilde{\nu}}{k_BT}\right) - 1\right)^{-1}.$$
 (B2)

Here,  $\tilde{\nu}$  has units of m<sup>-1</sup>,  $h=6.626\times10^{-34}$  J-s is Planck's constant,  $c=2.998\times10^8$  m-s<sup>-1</sup> is the speed of light,  $k_B=1.381\times10^{-23}$  J-K<sup>-1</sup> is Boltzmann's constant, and T is the water temperature in Kelvin.

- We use the spectral emissivity of calm water from Feldman et al. (2014) and the 1984-2008 median water temperature of 289 K at the lighthouse (https://www.ndbc.noaa.gov/data/climatic/CHLV2.txt) in Equations B2 and B1 to obtain  $\varepsilon_{wtr,0} = 0.9147$ . However, wind speed slightly increases the emissivity of water; the median climatological wind speed at the Chesapeake Lighthouse is 7.1 m-s<sup>-1</sup>, which adds  $\delta\varepsilon_{wtr,7} = 0.005$  to the calm water value (Huang et al., 2016). So the climatological-averaged broadband emissivity of water at the lighthouse is  $\varepsilon_{w} = \varepsilon_{wtr,0} + \delta\varepsilon_{wtr,7} = 0.9147 + 0.005 = 0.92$ .
- We note that the wind speed and water temperature variations at the lighthouse do not significantly alter the seawater emissivity. The climatological record (1984-2008) indicates that the minimum and maximum seawater temperatures recorded at the lighthouse are 273 and 302 Kelvin. This corresponds to calm water emissivities in the range of 0.912-0.916, with the lowest emissivities occurring at the lowest temperatures. The interquartile range of wind speeds span 5-10 m-s<sup>-1</sup>, and this adds 0.003-0.009 to the calm water emissivity (Huang et al., 2016). Since water temperature and wind speed are anti-correlated at the lighthouse (i.e., the coldest months are also the windiest months), the extreme range of emissivities is 0.919-0.921, or  $\varepsilon_w = 0.92 \pm 0.001$ .

## Appendix C: Detailed Description of the Component Summation Energy Balance

We presented the component summation flux in Equation 3 for local thermodynamic equilibrium conditions, which we restate here:

$$LW_{cs}^{\uparrow} \equiv LW_{t-0}^{\uparrow} = (1 - \varepsilon_1)[\varepsilon_w \sigma T_w^4 + (1 - \varepsilon_1)(1 - \varepsilon_w)LW^{\downarrow}] + \varepsilon_1 \sigma T_1^4. \tag{C1}$$

Importantly, this equation contains dimensionless transmission and reflectance coefficients as follows:

- $(1 \varepsilon_1)$  is the transmission of Layer 1 (dimensionless). Since  $\varepsilon_1 \simeq 0.015$  (see Section 4.2), the transmission of Layer 1 is  $(1 \varepsilon_1) \simeq 0.985$ , and nearly all radiation passes through this thin layer of the atmosphere.
  - $(1 \varepsilon_w)$  is reflection coefficient of the water (also dimensionless). Recall  $\varepsilon_w = 0.92$  (Appendix B), so the water reflectance is  $(1 \varepsilon_w) = 0.08$ . Thus, the atmospheric irradiance reflected off of the water is an order of magnitude less than the incident flux.

Now we can break down the component summation equation and assess the contribution of each term in Equation C1:

- The first term contains a familiar combination of variables,  $\varepsilon_w \sigma T_w^4$ , which is the irradiance emitted by the water. Recall  $\varepsilon_w = 0.92$  (Appendix B), so  $\varepsilon_w \sigma T_w^4 = 368.9 \text{ W-m}^{-2}$  when  $T_w = 290 \text{ K}$  (the average SST at COVE). Accounting for attenuation of radiation

as it travels through Layer 1 yields  $(1 - \varepsilon_1)\varepsilon_w \sigma T_w^4 = 363.4 \text{ W-m}^{-2}$  at the boom height. This water irradiance term dominates the component summation equations 3 and C1.

- The second term,  $(1 \varepsilon_1)^2 (1 \varepsilon_w) L W^{\downarrow}$ , represents the reflected downwelling flux after two-way transmission through Layer 1. Inserting values for  $\varepsilon_1$  and  $\varepsilon_w$  yields  $(1 \varepsilon_1)^2 (1 \varepsilon_w) L W^{\downarrow} = (0.985)^2 (0.08) L W^{\downarrow} \simeq (0.08) L W^{\downarrow}$ . The average  $L W^{\downarrow}$  at COVE is 339 W-m<sup>-2</sup>, so  $(1 \varepsilon_1)^2 (1 \varepsilon_w) L W^{\downarrow} = 26.3$  W-m<sup>-2</sup>. Thus, the second term is an order of magnitude smaller than the first term.
- The third term ( $\varepsilon_1 \sigma T_1^4$ ) represents the upwelling irradiance of the air that is below the sensors at COVE (i.e., Layer 1). Since  $\varepsilon_1 = 0.015$ , this term contributes 5.9 W-m<sup>-2</sup> at the average COVE surface temperature of  $T_1 = 289$  K. So the irradiance emitted by the air below the boom is two orders of magnitude smaller than the water emission given by the first term.
- Thus, the irradiance measured at the boom height is dominated by the water contribution (the first term). Downwelling atmospheric radiation that is reflected off of the water is an order of magnitude less than the irradiance emitted by the water, and the irradiance of the air between the boom and the water is two orders of magnitude less than the irradiance of the water below.

# Appendix D: Sensitivity of $\varepsilon_1$ to Atmospheric Emissivity

In Section 4.2 we presented an average  $\varepsilon_1 = 0.015$  for the layer of air below our sensors based upon Equation 7:  $\varepsilon_1 = 1 - (1 - \varepsilon_{atm})^{\eta}$ . We used a median scale factor of  $\eta = 0.011$  in this equation derived from four years of water vapor mixing ratios and total column precipitable water vapor measurements at the COVE site (2004-2007). Equation 7 also requires the emissivity of the atmosphere,  $\varepsilon_{atm}$ ; we used  $\varepsilon_{atm} = 0.75$  based upon values found in the literature, as explained in the next paragraph. In this section, we explore the sensitivity of  $\varepsilon_1$  to  $\varepsilon_{atm}$  and the effect of using  $\eta = 0.011$  instead of using the range of  $\eta$  determined at the COVE site (IQR = 0.009 to 0.014).

Finding a robust emissivity of the atmosphere in the literature was a challenge. Some authors attempted to determine the atmospheric emissivity with clear skies (Staley and Jurica, 1972) or with clear skies at night (Chen et al., 1991), while others chose to determine the emissivity of air in a variety of conditions (Sridhar and Elliot, 2002; Abramowitz et al., 2012; Kalinowska, 2019). We concluded that  $\varepsilon_{atm}$  can not be precisely specified because of changing atmospheric conditions (e.g., cloud cover), but that reasonable values of  $\varepsilon_{atm}$  range from  $\sim 0.6 - 0.9$  (Zhao et al., 2019). Thus, we chose  $\varepsilon_{atm} = 0.75$  and show the impact of using  $\varepsilon_{atm} = 0.6 - 0.9$  in Figure D1.

The median absolute differences in Figure D1 are near zero and the largest IQR is  $\sim 0.3$  W-m<sup>-2</sup> and occurs in winter when  $\varepsilon_{atm} = 0.9$ . 535 Clearly, the range of  $\eta$  and  $\varepsilon_{atm}$  measured at COVE have little effect on  $LW_{cs}^{\uparrow}$ .

Figure D1. This plot assesses the range of  $LW_{cs}^{\uparrow}$  values obtained using the median  $\eta = 0.011$  in Equation 7 or using the available data at the COVE site; this is done for three  $\varepsilon_{atm}$  values (0.6, 0.75, 0.9). The range of values increase as  $\varepsilon_{atm}$  increases (i.e., the whiskers get bigger), but the median biases are near zero.

Author contributions. Conceptualization: BEF, GLS, BL, FMD; Formal analysis: BEF, GLS, BL; Investigation/Methodology: BEF, GLS, FMD, BL, WS, ZAE; Supervision: GLS, BL, and DAR; Validation: BEF, GLS, BL; Visualization: BEF, GLS; Resources: NGL; Instrument maintenance: BEF, FMD and RFA; Data Curation: JJM; Writing (original draft preparation): BEF, GLS; BEF and GLS prepared the manuscript with significant contributions from BL, and other contributions from ZAE and DAR.

Competing interests. The authors declare that there are no competing interests.

Acknowledgements. We thank the United States Coast Guard (USCG) and particularly Lt. Ed Westfall, Lt. Frank Minopoli, and Carlos Hernandez (BOSN, ATON officer Sector Hampton Roads) for allowing NASA CERES research at the Chesapeake Light Station. Likewise, we also thank the Department of Energy (DOE), Jim Green and Rick Driscoll for access to the Chesapeake Light after ownership transferred from the Coast Guard to the DOE in 2012. Finally, this work could not have been possible without funding from the NASA CERES project.

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

Figure 1. Illustration of the instrument geometry at the Chesapeake Lighthouse.  $LW^{\uparrow}$  flux measurements are located on an 8 m boom, which is not long enough to remove the structure from the FOV.  $LW^{\downarrow}$  flux measurements have an unobstructed FOV at the top of the lighthouse structure tower. The Infrared Radiation Thermometer (IRT) provides sea surface temperature, and it has a narrow FOV that is unobstructed by the structure. The emissivities  $\varepsilon_{atm}$  and  $\varepsilon_1$  pertain to the atmospheric column and the layer of air below the boom.

**Figure 2.** Fish-eye photograph taken near the end of the boom of Figure 1. The picture shows the lighthouse legs/structure occupying a significant portion of the static pyrgeometer FOV.

Figure 3. Single year plot of the differences between the upwelling irradiances measured with a pyrgeometer  $(LW_{prg}^{\uparrow})$  and infrared radiation thermometer  $(LW_{IRT}^{\uparrow})$ . 445,578 points. The upper portion of the plot (sza > 90) is nighttime data and the lower portion (sza 

Figure 4. Similar dataset to Figure 3, but plot of the differences between the component summation longwave irradiance measurements defined in Equation 3 ( $LW_{cs}^{\uparrow}$ ) and the IRT measurements. Notice the data is now symmetrical throughout the course of a day without the influence of the structure. There are still large biases between the summer and winter and this is due to the reflectance term being higher in the summer.

Figure 5. Longwave upwelling irradiances measured on two winter days; left panel is a clear day and right panel is an overcast day. Black lines: Pyrgeometer  $(LW_{prg}^{\uparrow})$ . Light blue: Infrared Thermometer  $(LW_{IRT}^{\uparrow})$ . White lines: Longwave component summation calculated using Equation 3 with  $\varepsilon_1 = 0.015$   $(LW_{cs}^{\uparrow})$ ; this measurement is the most accurate of all irradiances in the figure. Red line: Air temperature (in Kelvin on the right Y-axis). Red squares: Water temperature (in Kelvin on the right Y-axis). Yellow shading: Solar elevation (not to scale). The pyrgeometer measurements (black line) are biased low of the longwave component summation measurements (white lines) because the cold lighthouse obstructs the field of view of the warm water.

Figure 6. Longwave upwelling irradiances measured on two summer days; left panel is a clear day and right panel is an overcast day. The color schemes are the same as the winter days shown in Figure 5. The anomaly between the pyrgeometer measurements (black line) and the longwave component summation measurements (white lines) increases throughout the sunny summer day, indicating that the pyrgeometer is capturing the direct solar heating of the lighthouse structure. This direct solar heating does not occur on the overcast day, but the pyrgeometer does capture a blast of warm air that heats the lighthouse structure at sunset. Note that  $LW_{cs}^{\uparrow}$  is minimally affected by air temperature in Figure 5 and Figure 6 because the water emission term dominates Equation 3; additional discussion is provided in Section 5.2

Figure 7. Monthly (left panel) and yearly (middle panel) boxplots of relative biases of the pyrgeometer measurements  $(LW_{prg}^{\uparrow})$  with respect to the component summation measurements  $(LW_{cs}^{\uparrow})$  over a 10 year period; the detached boxplot on the far right applies to all 10 years. The shaded region represents the BSRN targeted accuracy of  $\pm 2\%$ . The top X-axis shows the percentage of data that fall outside the target accuracy. Boxes capture the interquartile range (middle 50-percentile) and red lines represent medians. Notches (barely perceptible) indicate the uncertainty of the medians with an approximate 95% confidence level. The bottom bar plot in the Yearly Variability section illustrates the median annual difference between air temperature and SST. Outliers are removed for visualization (defined as values that are more than 1.5 times the interquartile range) and whiskers represent the non-outlier minimums and maximums. Minimum number of points per box is 5505 for monthly boxplots and 4873 for yearly boxplots.

Figure 8. Computed relative biases associated with structures that obstruct different percentages of the field of view for hemispherical long-wave measurements. Computations are based upon Equation 12 in Section 6.1. At COVE, the pyrgeometer had  $\sim 15\%$  structure obstruction. The pyrgeometer with 5% structure obstruction would be within BSRN target uncertainty. This would have required extending the boom at COVE an additional 6 m (14 m total) from where it was located.