# Peer review of "The Component Summation Technique for Measuring Upwelling Longwave Irradiance in the Presence of an Obstruction"

_EGUsphere, 2025_

## Author Comment (AC1)

**Response to Reviewer 1**

Reviewer comments in *italic.*

*Overall, the paper discusses a highly important alternative to determining longwave irradiance if an obstruction is present. This change will help improve data collection at environmental monitoring stations, both current and that may be in use in the future. I recommend this article for publication, following some minor revisions outlined below.*

Hello, thank you for taking the time to review this paper, recommending the article for publication and for your suggestions. We believe this has made the paper stronger and more detailed. To respond to your suggestions:

***Line 28: Please remove the acronym definition of NASA from Line 32 to Line 28. This is the first place it is mentioned.***

We made the change from Line 32 to 29 (former line 28).

***Equation 2: Is this a derived equation? Or created by the authors as a model for collected data? A reader would benefit from having some background on the established relationships and source of all equations, not just Equation 2.***

Thank you for pointing this out as we did not make this clear. That equation is a LW energy balance equation that is similar to the one presented by Lee et al. (2010), (page 24), but we also include the emissivity of air. We have re-written the introduction to Eq 3 (formerly Eq 2) to aid readers:

> "The air temperature and skin temperature of the water within the small field of view of the down-looking measurements at COVE are very likely isothermal, so the upwelling radiation at COVE is isotropic (Stephens, 1994, Section 7.1). Since the air temperature and water skin temperature are horizontally homogeneous, we can assume that they are both gray bodies and use the Stefan-Boltzmann law to relate Planck's function to broadband irradiance. Hence, we can use an energy balance (e.g., Lee et al., 2010) to express $LW_{f=0}^{\uparrow}$ as a component sum of measurements that do not require a pyrgeometer and are not influenced by the lighthouse:"

We also added Appendix C, which breaks down Equation 3 (formerly Eq. 2) to assist the reader in understanding the contributions of each term.

A review of the equations were conducted and sourced where needed. We explained how to compute the fraction of blocked upwelling ($f$) in a new Appendix A. This will hopefully provide enough background information for the reader.

*Line 260—262, related to Figures 6 and 7: is the expectation that the LWcs is to be pretty consistent and not impacted by air temperatures? Mentioning that the adjusted data matches expectations (and why) could help emphasize the validity of the new measurement technique.*

Yes. The expectation for Fig. 5 and 6; Lines 266-268(formerly Fig. 6 and 7; former Line 262-264) is that LWcs is dominated by the water temperature. This is because the terms in Equations 3 that involve air temperature are small compared to the water emission. We did not emphasize this enough in the paper, so we've added the following paragraph to the beginning of Section 5.2 (Former section 5.3):

"In this section we analyze four single-day scenarios in winter and summer and in clear and overcast conditions to help understand the physics driving the biases in Figures 3 and 4. Since the water emission term in Equation 3 ($\varepsilon_w \sigma T^4$) dominates the radiative flux (see Appendix C), we expect the true upwelling flux to track the IRT measurements throughout the day (albeit with a high bias) and only minor perturbations associated with the air temperature. In the following paragraphs, though, we see that the pyrgeometer measurements can be significantly affected by changes in air temperature and/or solar heating of the lighthouse structure."

*Line 280: How is FOV found/calculated? This explanation may help readers trying to apply this to their own situation.*

Thank-you for bringing this up – Reviewer 2 also mentioned this, so we use similar text for both responses.

The parameter $f$ is the fraction of flux that is blocked by the lighthouse, and it was first introduced after Eq 1. This was not stated clearly in the original draft, so we have modified the text after Equation 1 to state:

"...where $f$ is the fraction of the upwelling irradiance that is blocked by the structure, $LW^{\uparrow}_{f=0}$ is the upwelling longwave flux in the absence of the structure, and $LW^{\uparrow}_{twr}$ is the upwelling longwave flux emitted by the structure."

We have also added Appendix A to describe the physical basis for computing flux in the presence of an obstruction with a simple shape (a rectangular cuboid). The appendix concludes with some text about how we use ImageJ software to estimate $f$ for the complex shape of the lighthouse, and we acknowledge that our $f$ value is our best estimate:

> "The geometry of the Chesapeake Lighthouse is more complicated than a rectangular cuboid, so we used imaging software to obtain $f \simeq 0.15 \pm 0.05$ for the lighthouse. Note that $f$ is not needed for the component summation technique (Section 4.1, Equations 3). However, $f$ is helpful for estimating the radiation perturbation caused by obstructions at other sites (Section 6.1)"

Finally, we added text to the beginning of Section 6.1 that describes briefly describes how we use the ImageJ software and our estimated accuracy of $f$.

> "Thus far, we have presented results specific to the geometry of the COVE platform, which obstructs about 15% of the pyrgeometer upwelling measurement (i.e., $f \simeq 0.15$). The obstruction percentage was calculated using a software package called ImageJ (https://imagej.net/ij/), which provides area and pixel value calculations within manually selected regions. In our application, we used ImageJ to distinguish the structure from the water surface and to estimate the percentage of the upwelling occupied by the structure. We conservatively estimate the accuracy of $f$ as $\pm 0.05$ because the camera used for Figure 2 is not precisely positioned at the pyrgeometer location and it may not be exactly level. Additional discussion about how $f$ is affected by an obstruction's geometry is provided in Appendix A."

*Figure 8: The x-axis should have a lower-case F to match the variable in Equation 11.*

Thank you for catching this detail. The Fig. 8 plot was changed to match the variable in Eq. 11 and to match lower case f throughout the article.

**References**

Lee, H.-T., Laszlo, I., and Gruber, A.: ABI Earth Radiation Budget Upward Longwave Radiation: Surface (ULR) ATBD, Tech. Rep. Version 2.0, NOAA NESDIS Center for Satellite Applications and Research, 2010.

---

## Author Comment (AC2)

**Response to Reviewer 2**

Reviewer comments in *italic*.

*Review of "The Component Summation Technique for Measuring Upwelling Longwave Irradiance in the Presence of an Obstruction" by Fabbri et al.*

*The paper introduces a long-wave component-summation technique to obtain accurate upwelling long-wave (LW_up) irradiance when a support structure partially blocks a pyrgeometer's field-of-view, a problem encountered at NASA's Chesapeake Light Station (COVE) surface validation site. The method reconstructs unobstructed LW_up by combining (i) sea-surface emission from an infrared radiation thermometer, (ii) reflected downwelling LW from a standard pyrgeometer, and (iii) emission from the thin air layer between the water and the sensor. Applying the approach to ten years of COVE data (2004–2013) shows that conventional pyrgeometer readings suffer a systematic bias, largely outside the Baseline Surface Radiation Network (BSRN) target of ±2%. The component-summed fluxes remove these seasonal biases and recover a long, homogeneous LW_up record suitable for Clouds and the Earth's Radiant Energy System (CERES) satellite validation. Sensitivity tests indicate that if an obstruction occupies less than ∼5% of the sensor's hemispherical view, residual errors remain below the BSRN threshold 99% of the time, offering a clear siting guideline.*

*Overall, this manuscript is well organized and easy to follow. However, the topic is quite narrow and highly specialized; the manuscript reads more like a technical support note than a broadly scoped research article. Several technical details—particularly the justification of key assumptions and the sensitivity of the component terms—should be clarified and documented in greater depth. In light of these considerations, I recommend major revision. More specific questions and comments are provided below.*

Thank you for reviewing this manuscript and providing valuable suggestsions that we hope are answered below. We believe this has made the paper stronger and more descriptive to the reader.

**1. Why is the pyrgeometer measurement even needed?**
*Throughout the manuscript the IRT-derived upwelling long-wave irradiance (LW_IRT) is treated as the reference against which both the obstructed pyrgeometer and the component-summation reconstruction are evaluated. If the IRT alone already delivers an accurate, obstruction-free LW_up measurement, what additional scientific or operational value does the pyrgeometer provide? Please clarify in the Introduction or Discussion why a long-wave pyrgeometer record remains necessary (e.g., calibration traceability, stability for climate time series, complementary performance in cloudy or high-humidity conditions, logistical redundancy, cost, or broader site comparability). Addressing this point will better motivate the need for the component-summation technique.*

We presented the IRT as an instrument that measures radiation that is unaffected by the

lighthouse, but we do not mean to represent IRT measurements as the "truth." Figure 4 demonstrates that the IRT-based upwelling is biased low of the component-summation method; this is mainly because the IRT does not account for the longwave downwelling reflected off the water (although the longwave emitted from the air below the COVE instruments also makes a small contribution). We present the relative contribution of all of the terms of Equation 3 (former Eq. 2) in the new Appendix C, where we see that the contribution of the reflected LW downwelling is about 10% of the LW upwelling. Thus, IRT measurements alone can not provide $LW^\uparrow$.

We also note that the pyrgeometer is a broadband instrument that measures LW directly (both upwelling and downwelling, depending upon orientation) without having to characterize the emissivity of the surface or the air, but they do not always work properly and it can be difficult to know when they are malfunctioning. Thus, we suggested that the longwave component summation (LWCS) method could be used as a check for the pyrgeometer measurements in the Discussion (Section 6.2, first 2-3 paragraphs). Basically, if the difference between LWCS and the pyrgeometer measurements are too large (as in Figure 4) or show diurnal variability (as in Figure 3), then there is an inconsistency between the techniques that can be caused by a variety of factors (for example, poor calibration or the perturbation of the radiation field by an obstruction). The availability of two techniques for measuring LW upwelling (i.e., pyrgeometer and component summation) allows users to establish exactly when the two measurement methods diverge and to troubleshoot the cause in a timely manner.

We added more text at the end of the Discussion section to explain why an unobstructed pyrgeometer measurement has its advantages over the component summation technique:

> "While our newly developed component summation approach provides an alternative method for obtaining $LW^\uparrow$, pyrgeometers offer operational simplicity via a single direct irradiance measurement (instead of using a combination of measurements) and remain the preferred option in unobstructed environments. This is because pyrgeometers account for reflected $LW^\downarrow$ without requiring knowledge of the broadband emissivity for the surface or the air below the sensors. This is especially important over land, where surface emissivity can vary with seasonal vegetation changes, surface moisture, and drought. Hence, measurement networks like BSRN recommend pyrgeometers for longwave irradiance measurements, but note that BSRN will accept $LW_{cs}^\uparrow$ as an ancillary approach."

**2. Clarification of the physical basis for the component-summation method (Eq. 2).**

*Equation (2) is the pivotal relation in the manuscript, yet its right-hand side is presented in compact form and the physical meaning of each term is only briefly touched upon. I recommend rewriting Eq. (2) so that the three component terms are shown explicitly—ordered from largest to smallest contribution—and providing a short paragraph that explains the origin, units, and expected magnitude of each term (sea-surface emission, reflected downwelling LW,*

*near-surface air-layer emission). Doing so will help readers grasp both the physical logic and the relative importance of the terms.*

Thank-you for this suggestion – Reviewer 1 had the same concerns, so we repeat that anwser here:

That equation is a LW energy balance equation that is similar to the one presented by Lee et al. (2010), (page 24), but we also include the emissivity of air. We have re-written the introduction to Eq 3 (formerly Eq 2) to aid readers:

> "The air temperature and skin temperature of the water within the small field of view of the down-looking measurements at COVE are very likely isothermal, so the upwelling radiation at COVE is isotropic (Stephens, 1994, Section 7.1). Since the air temperature and water skin temperature are horizontally homogeneous, we can assume that they are both gray bodies and use the Stefan-Boltzmann law to relate Planck's function to broadband irradiance. Hence, we can use an energy balance (e.g., Lee et al., 2010) to express $LW^{\uparrow}_{f=0}$ as a component sum of measurements that do not require a pyrgeometer and are not influenced by the lighthouse:"

We also added Appendix C, which breaks down Equation 3 (formerly Eq. 2) to assist the reader in understanding the contributions of each term.

**3. Assumptions and limitations of Eq. (1).**
*Eq. (1) is written as if it were generally valid, yet for a hemispheric pyrgeometer the effect of an obstruction depends not only on the fractional blockage f but also on its position within the field-of-view (FOV). For the same f, an obstruction near zenith (nadir-viewing for LW ↑) will bias the measurement far more than one near the horizon. Please (i) state the physical basis of Eq. (1), (ii) discuss whether the derivation assumes uniform angular radiance, and (iii) clarify how, or whether, blockage geometry is accounted for in the 10-year COVE application.*

Our apologies for the imprecise description of Equation 1, and thank-you for bringing this to our attention. The parameter $f$ refers to the fraction of flux that is blocked, which we did not state clearly in the original draft. We have modified the text after Equation 1 to state

"...where $f$ is the fraction of the upwelling irradiance that is blocked by the structure, $LW^{\uparrow}_{f=0}$ is the upwelling longwave flux in the absence of the structure, and $LW^{\uparrow}_{twr}$ is the upwelling longwave flux emitted by the structure."

We have also added Appendix A to describe the physical basis for computing flux in the presence of an obstruction with a simple shape (a rectangular cuboid). The appendix concludes with some text about how we use ImageJ software to estimate $f$ for the complex shape of the lighthouse.

> "The geometry of the Chesapeake Lighthouse is more complicated than a rectangular cuboid, so we used imaging software to obtain $f \simeq 0.15 \pm 0.05$ for the lighthouse. Note that $f$ is not needed for the component summation technique (Section 4.1, Equation 3). However, $f$ is helpful for estimating the radiation perturbation caused by obstructions at other sites (Section 6.1)"

Finally, we added text to the beginning of Section 6.1 that describes briefly describes how we use the ImageJ software and our estimated accuracy of $f$.

> "Thus far, we have presented results specific to the geometry of the COVE platform, which obstructs about 15% of the pyrgeometer upwelling measurement (i.e., $f \simeq 0.15$). The obstruction percentage was calculated using a software package called ImageJ (https://imagej.net/ij/), which provides area and pixel value calculations within manually selected regions. In our application, we used ImageJ to distinguish the structure from the water surface and to estimate the percentage of the upwelling occupied by the structure. We conservatively estimate the accuracy of $f$ as $\pm 0.05$ because the camera used for Figure 2 is not precisely positioned at the pyrgeometer location and it may not be exactly level. Additional discussion about how $f$ is affected by an obstruction's geometry is provided in Appendix A."

**4. Treatment of angular and spectral variation.**
*Both Eq. (1) and Eq. (2) ignore the angular and spectral dependence of the LW radiation. But in reality, upwelling LW radiance varies with zenith angle and wavelength. While a gray, isotropic assumption may be defensible for this specific problem, the manuscript should (a) acknowledge the simplification, (b) justify its adequacy (e.g., by order-of-magnitude estimates or citing prior studies), and (c) discuss how angular or spectral departures from gray-Lambertian conditions might influence the stated $\pm 2\%$ accuracy target*

Very interesting point... we did assume a plane-parallel and isotropic atmosphere, as this is the universal textbook approach for modeling longwave irradiance in the atmosphere. We did not explain why or the validity of the approach, though, which is important. Thank-you for catching this.

In our case, we only need to consider two layers: the optically thin layer of the atmosphere located below the boom height ($\sim$21 m) and the ocean surface layer. We assume that both layers are isothermal within the field of view of the COVE instruments (which is very reasonable, since the Chesapeake Lighthouse is surrounded by the Atlantic Ocean, and the nearest land is 25 km away). Since we are assuming that Layer 1 has a constant temperature everywhere within the sensors field of view (horizontally and vertically) and that the ocean skin temperature is also constant, the Planck function does not vary within the layers and the radiation field that the down-looking COVE sensors observe is indeed isotropic. Unlike downwelling, upwelling radiation from an isothermal atmosphere does not

exhibit limb brightening and the radiation field is constant with respect to zenith angle (e.g., Stephens, 1994, Equation 7.8). We added text to cover this at the beginning of Section 4.1 by stating:

> "The air temperature and skin temperature of the water within the small field of view of the down-looking measurements at COVE are very likely isothermal, so the upwelling radiation at COVE is isotropic (Stephens, 1994, Section 7.1). Since the air temperature and water skin temperature are horizontally homogeneous, we can assume that they are both gray bodies and use the Stefan-Boltzmann law to relate Planck's function to broadband irradiance. Hence, we can use an energy balance (e.g., Lee et al., 2010) to express $LW_{f=0}^{\uparrow}$ as a component sum of measurements that do not require a pyrgeometer and are not influenced by the lighthouse:"

Regarding wavelength dependence, the pyrgeometer does not spectrally resolve the irradiance field, so we used broadband emissivities throughout the paper. We showed how we computed the broadband emissivity of water in Appendix B, and the emissivities that we provided for the atmosphere also apply to the broadband (although we leaned on the scientific literature for these values). However, this was not stated clearly in the paper, so we added the following text to the introduction of Section 4 (around Line 143):

> "We have assumed that objects emit negligible radiation outside of the spectral range of the pyrgeometer at terrestrial temperatures throughout this paper. Thus, we have approximated the Stefan-Boltzmann law as
>
> $$\int_{4\,\mu\mathrm{m}}^{50\,\mu\mathrm{m}} \varepsilon(\lambda)B(T,\lambda)d\lambda \simeq \int_{0}^{\infty} \varepsilon(\lambda)B(T,\lambda)d\lambda = \frac{\varepsilon_b \sigma T^4}{\pi}, \tag{1}$$
>
> where $\varepsilon(\lambda)$ is the emissivity at wavelength $\lambda$, $B(T,\lambda)$ is the Planck function at temperature T and wavelength $\lambda$, and $\varepsilon_b$ is the broadband emissivity. We demonstrate how to compute the broadband emissivity of water in Appendix B."

***5. Line 384: Plank's constant should be Planck's constant***
Thank-you for catching this. It has been changed on its new line (line 488).

**References**

Lee, H.-T., Laszlo, I., and Gruber, A.: ABI Earth Radiation Budget Upward Longwave Radiation: Surface (ULR) ATBD, Tech. Rep. Version 2.0, NOAA NESDIS Center for Satellite Applications and Research, 2010.

Stephens, G.: Remote sensing of the lower atmosphere, Oxford University Press, 1994.